# Comment on Laming et al. The Curse of Conservation: Empirical Evidence Demonstrating That Changes in Land-Use Legislation Drove Catastrophic Bushfires in Southeast Australia. *Fire* 2022, *5*, 175

## Michael Charles Feller

Department of Forest and Conservation Sciences, University of British Columbia, Vancouver, BC V6T 1Z4, Canada; michael.feller@ubc.ca

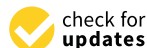



Laming et al. [1] state that their results "implicate land-use legislation changes as the root cause for catastrophic bushfires in the Buchan region, and likely across southeast Australia" (P. 14). Their paper presents no sound evidence to support this. The reasoning used to support this statement is faulty and contains incorrect information.

Thus:

1. Throughout the paper, the authors make erroneous comments about the Victorian Land Conservation Council (LCC) Act of 1970, claiming that the LCC Act aimed to ban prescribed burning. Absolutely nowhere in the Act is there any reference to prescribed burning or its banning. Furthermore, the LCC made no recommendations about the use of fire on private land.

2. The study area of Laming et al. occurs just inside the Gippsland Lakes Hinterland region of the LCC, with the LCC East Gippsland region a short distance to the east. The Final Recommendations of the LCC for both these areas [2,3] allowed for prescribed burning on both protected and unprotected public land. Nowhere did the LCC recommend bans on prescribed burning, contrary to the assertions made in Laming et al.

3. Laming et al. found an increase in charcoal and a change in vegetation during the 1970s in their sediment core. They attribute this to a shift from high-frequency, low-intensity fires lit by landowners to lower-frequency, high-intensity fires when landowners were prevented from burning by the LCC around 1970. This cannot be true because the LCC had no impact on landowners around the authors' study area until 1979–1984, when the LCC recommendations were implemented. Laming et al. quote a single local farmer who stated that the LCC "actively prohibited settler mimicry burning" "in about 1970" (P. 6) but have no real evidence to support their assertion. Furthermore, settler burning did not mimic indigenous burning, as found in several studies, including Gell et al. [4] and Wakefield [5].

4. The authors' interpretation of their charcoal and pollen results is entirely speculative and faulty, is not supported by the published literature, and does not consider known important causes of fires.

Firstly, Laming et al. have no data to indicate whether the charcoal came from more fires, such as campfires, closer to their study site or from more severe and widespread fires.

Secondly, Laming et al. provide no data to rule out the most important drivers of catastrophic bushfires. Climate change is critically important in determining fire behaviour. It has been well established that weather, rather than fuel, is the dominant factor controlling high-intensity fire [6–9]. Clark et al. [10] found increased temperature to be the main driver of heightened fire danger in Victoria, Australia.

Laming et al. cannot eliminate climate change as the major cause of charcoal changes as they provide no analysis of climate impacts on fires in their area, such as the incidence

of high fire danger days over time, which have been found to increase in recent years in Tasmania [11].

As people are the major causes of fires in the region and the study area is close to a popular recreation site, the authors also cannot eliminate changes in human visitation or use in the area as the major causes of charcoal changes. Collins et al. [12] concluded that people are the main cause of wildfire in SE Australia and that increased fires will occur due to population increases and climate change effects. Penman et al. [13] found that climate change and population increase were the main drivers of increases in ignition frequency in their study area in NSW.

Timber harvesting has been common in the region around the author's study area. Timber harvesting has also led to increases in high-intensity fires in the region [14]. Laming et al. also ignore this possible contribution to the fire regime.

5. Laming et al. extrapolate data for their one study site to the entire region. This cannot be done, particularly when a previous study of pollen and charcoal in sediment elsewhere in East Gippsland produced completely different conclusions. Gell et al. [4] found only low levels of burning in their area prior to European contact. Thereafter, fire incidence increased due to burning by settlers, miners and timber cutters. This resulted in a change in the understorey vegetation. This vegetation reverted to its shrubbier, pre-European-contact type following post-1939 fire suppression activities. Although this study used methods that have subsequently been superseded, Gell (2023. personal communication) considered that the method of using the % eucalyptus as a denominator at the time was valid given the reliability of forest cover over those time frames (and given that for 300+ years, trees dominated the catchment). So, the Gell et al. charcoal record is correct unless eucalyptus pollen changed by orders of magnitude over 200 years, which was not possible given that most Eucalyptus pollen is wind-transported and the site sits in the midst of the most forested part of Victoria.

The assertions of Laming et al. are also contradicted by more recent charcoal/paleofire studies [15,16] elsewhere in SE Australia which indicate climate and increased human ignitions have been the main drivers of the fire regime.

Laming et al. present no data on pollen and charcoal prior to 1910, so they cannot know that their earlier data simply reflect enhanced burning post European contact and that the most recent vegetation is similar to what was present prior to such contact.

Furthermore, in a comprehensive study of indigenous burning in Australia, Williams et al. [17] concluded that " . . . human control of fire by prehistoric people in Australia is not evident at broad landscape levels". There might have been some local human control of fire, but it was not widespread. Laming et al. ignore this study, and their caption for their Figure 7, which states that " . . . prior to British Invasion, Indigenous people managed variable cultural landscapes with fire across all parts of the region from open grasslands to closed forests in southeast Australia", is unsubstantiated and contradicted by considerable research, including Williams et al. [17] and Lindenmayer and Bowd [18].

**Acknowledgments:** I thank three anonymous reviewers for their helpful comments on an early draft of this manuscript.

**Conflicts of Interest:** The author declares no conflict of interest.

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
