# Peer review of "Comment on Laming et al. The Curse of Conservation: Empirical Evidence Demonstrating That Changes in Land-Use Legislation Drove Catastrophic Bushfires in Southeast Australia. Fire 2022, 5, 175"

_fire, doi:10.3390/fire6070260_

Round 1

Reviewer 1 Report

The work by Laming et al presents several problems, and it is interesting that you highlight some issues (e.g. I dove into the Land Conserv Act 1970 and Land Conserv Council docs referenced in Laming et al and was shocked to see that they don't mention fire!).

So: I only saw a few things, mostly minor.

1. the reference to Gell et al. (line 61) is OK given its proximity to the site used by Laming et al but that work is old and uses methods that have largely been superseded. You might just consider tempering your language when using that example.

2. line 64 you have "understorey", perhaps US spelling, whereas I use "understorey".

3. line 70 there is an "s" at the start of the Williams et al (ref 13) quote, perhaps misplaced.

4. line 73 probably should be capital F when referring to the Figure?

5. In you concluding statement (line 76) you state that their work is unsubstantiated and contradicted by William et al (ref 13). I believe that statement could be broadened e.g. 'unsubstantiated and contradicted by considerable research including Williams et al. [13].'

I also note that many of your charcoal/palaeofire refs (refs 12 and 13) are a little old... there is more recent work including in Fire (e.g. https://doi.org/10.3390/fire6040152) and elsewhere (e.g. https://doi.org/10.1177/09596836231151827)

Reviewer 2 Report

This is an excellent short response to the Laming et al. paper in Fire. 

I have read the text several times - my only comment is that on Line 40 the authors are discussing climate change. The word "Thus" opens the sentence. Please remove the word because one does not follow another. Because Laming did not examine climate change does not necessarily mean it is important in this context. 

My strong opinion is that Laming et al. should never have been published in the first place - and the critiques of the paper clearly show why

Author Response

My response to comments of reviewer 2 are:

  1. Use of "Thus" : I have removed the word (L 42)
  2. Laming et al. should never have been published: I fully agree and consider that the paper should be withdrawn in the interests of scientific integrity.

Reviewer 3 Report

This paper addresses some critical points about an influential study, and I think it makes an important contribution. There are some issues that need to be addressed, although they do not affect the conclusions of the author.

Lines 28-30 make the important point that the LCC recommendations were not implemented until after the time that Laming et al claim they had an effect. Confusingly, however, the author uses the phrase "and their burning", after having stated earlier that the LCC did not affect burning. To my knowledge the earlier statement is correct, so I suggest removal of "and their burning".

Lines 30-32: The author makes the critical point that the sole basis for the argument from Laming et al that the act stopped burning and resulted in massive increase in fuels comes from the opinion of a single individual, not from recorded evidence. He also demonstrates that this conflicts with the historical record, so the individual is in error. I suggest removing the speculation about the individual's memory.

Lines 59-63: This evidence directly conflicts with the central claim of "settler mimicry burning" made by Laming et al. In contrast to Laming et al, Gell et al compared "settler" burning to Aboriginal burning and showed that it was vastly different. I suggest the author clarify the point here that the claim of "settler mimicry burning" is a disproved assertion.
